# Functionalization of Ordered Mesoporous Silica (MCM-48) with Task-Specific Ionic Liquid for Enhanced Carbon Capture

**DOI:** 10.3390/nano14060514

**Published:** 2024-03-13

**Authors:** Firuz A. Philip, Amr Henni

**Affiliations:** Faculty of Engineering and Applied Science, University of Regina, Regina, SK S4S 0A2, Canada; philip2f@uregina.ca

**Keywords:** CO_2_ capture, mesoporous silica, MCM-48, amino acid ionic liquid, task-specific ionic liquid

## Abstract

This work presents new composites of AAILs@MCM-48 obtained by functionalizing ordered mesoporous silica MCM-48 with two amino acid-based ionic liquids (AAILs) ([Emim][Gly] and [Emim][Ala]) to improve carbon capture and the selectivity of CO_2_ over nitrogen. Thermogravimetric and XRD analyses of the composites showed that the MCM-48 support’s thermal and structural integrity was preserved after the AAILs were encapsulated. An N_2_ adsorption–desorption study at 77 K confirmed AAIL encapsulation in the porous support. Under post-combustion flue gas conditions, both [Emim][Gly]@MCM-48 and [Emim][Ala]@MCM-48 demonstrated improved CO_2_ adsorption in comparison to the unmodified MCM-48, with a CO_2_ partial pressure of around 0.15 bar. Regarding the maximal CO_2_ uptake, the 40 wt.%-[Emim][Gly] composite outperformed the others at 303 K, with values of 0.74 and 0.82 mmol g^−1^, respectively, at 0.1 and 0.2 bar. These numbers show a 10× and 5× increase, respectively, compared to the pure MCM-48 under identical conditions. In addition, the selectivity of the composites was improved significantly at 0.1 bar: the selectivity of composites containing 40 wt.% [Emim][Ala] increased to 17, compared to 2 for pristine MCM-48. These composites outperform other silica-based studies reported in the literature, even those using amines as solvents. The presented composites offer therefore promising prospects for advancing carbon capture technology.

## 1. Introduction

Carbon capture and storage (CCS) technology is of paramount importance in global efforts to mitigate climate change by reducing carbon dioxide (CO_2_) emissions from industrial processes and power generation. According to the International Energy Agency (IEA), CCS has the potential to contribute significantly to reducing global CO_2_ emissions by up to 8% by 2050 [1]. Furthermore, the Intergovernmental Panel on Climate Change (IPCC) highlights the critical role of CCS in limiting global warming to 1.5 °C, emphasizing its necessity in achieving international climate targets [2]. As nations strive to transition to more sustainable energy practices, CCS emerges as a vital technology for capturing and storing CO_2_ emissions, thereby mitigating the impacts of climate change.

Among the various carbon capture technologies, solid sorbents have emerged as promising candidates. Silica-based solid sorbents like SBA-15, KIT-6, MCM-41, and MCM-48 have garnered a lot of interest owing to their high surface areas and large pore volumes, in addition to their great thermal and chemical stabilities [3,4,5,6]. Despite this, their use in the CO_2_ capture process has been restricted owing to their poor CO_2_ capture capabilities, which are caused by a lack of affinity toward CO_2_. This is particularly true in post-combustion settings, where the partial pressure of CO_2_ is close to 0.15 atm. Other research groups have used procedures that include functionalizing the pore surface with amines or ionic liquids to improve their capacity for CO_2_ adsorption [7,8,9,10]. MCM-41 is an ordered mesoporous material that was used by some researchers for the purpose of surface functionalization [9,11,12]. MCM-41 has a one-dimensional pore channel and is sensitive to the restricted diffusion of guest molecules as well as pore obstruction. Alternatively, MCM-48 has a three-dimensional cubic pore structure. MCM-48 is therefore a more desirable option than MCM-41 as it provides a better diffusion channel for the guest molecules and is less likely to become blocked [3,13].

Ionic liquids (ILs), attractive for their low volatility and high thermal stability, have been proposed as a promising alternative to amines in CO_2_ capture. However, while some studies have shown improved selectivity with IL-modified sorbents, there are instances where CO_2_ uptake diminishes [14,15,16]. Bates et al. [17] pioneered the development of amine-functionalized ionic liquids (AILs) which significantly enhanced CO_2_ sorption while preserving IL properties. These AILs, often termed task-specific ionic liquids (TSILs), have seen widespread adoption in scientific research. Among many reported TSILs, amino acid anion-functionalized ionic liquids (AAILs) offer notable advantages including easy synthesis, low cost, biodegradability, and eco-friendly sourcing from naturally occurring amino acids [18,19,20]. Fu et al. demonstrated that AAILs such as [Bmim][Gly] can act as promoters and contribute to a high absorption rate and large absorption capacity [21,22,23]. AAILs with an amine functional group have a greater capacity to capture CO_2_ compared to other physical ILs, and hence are more suitable candidates for immobilization within the solid support. Wang and co-researchers [24] showed that by encapsulating [Emim][Gly] and [Emim][Ala] inside the nanoporous polymethylmethacrylate (PMMA) microspheres, they were able to enhance the absorption of CO_2_ and obtain faster CO_2_ reaction rates. In our previous work, we have also demonstrated an enhanced CO_2_ capture capacity and CO_2_ selectivity through immobilization of ([Emim][Gly] and [Emim][Ala] into ZIF-8 [25] and MOF-177 [26].

Hence, for this investigation, we chose these two amino acid ionic liquids to be immobilized into an ordered mesoporous silica, MCM-48. The goal was to create AAIL@MCM-48 composites that can enhance CO_2_ adsorption and increase its selectivity over N_2_. To our knowledge, no research has been published addressing functionalized MCM-48 with these two TSILs. Our study focused then on characterizing and investigating the composites of AAILs@MCM-48 from an engineering standpoint. Specifically, this study examined their CO_2_ capture capacity, selectivity, and enthalpy of adsorption, and modeled the adsorption isotherms. The CO_2_ adsorption isotherm was determined at three distinct temperatures (303, 313, and 323 K) and for pressures ranging from 0.10 to 10.0 bar. Furthermore, nitrogen adsorption isotherms were evaluated for both composites and the original MCM-48. This allowed for the calculation of the optimum selectivity of CO_2_/N_2_.

## 2. Materials and Methods

### 2.1. Materials

The chemicals used were methanol, [Emim][Gly] (CAS: 766537-74-0), [Emim][Ala] (CAS: 766537-81-9), and MCM-48 (CAS: 7631-86-9), and they were acquired from Sigma Aldrich (Oakville, ON, Canada). The structure of the cation and anion of the amino acid ionic liquids (AAILs) are presented in Table 1. Before the preparation of the sample, MCM-48 was dried at 150 °C. The TSILs and MCM-48 and their composite samples were stored inside a glovebox (Clean Tech LLC, Orange, CA, USA)) under an inert atmosphere with flowing argon gas to limit moisture and CO_2_ adsorption. CO_2_ (99.99 vol. %) and N_2_ (99.99 vol. %) used in the adsorption experiments were acquired from Praxair Inc., Mississauga, ON, Canada.

### 2.2. Preparation of AAIL@MCM-48 Composite

The MCM-48 silica support was functionalized with the AAILs using the classical wet impregnation method, with the assistance of methanol [26]. To achieve this, the desired amount of AAILs was mixed with 5 mL of methanol and shaken for 30 min to homogenize the solution. The resulting solution was then added dropwise to the preweighed MCM-48 in a separate vial and shaken for 1 h. After 24 h of ambient solvent evaporation, any remaining solvent was removed by drying the composite at 80 °C for 2 h. For each AAIL, three different composites were prepared by varying the content of AAIL (20, 30, and 40 wt.%). The composites obtained were then labeled as X-AAIL@MCM-48, where X denotes the weight percentage of AAILs employed. For example, the composite 20-[Emim][Gly]@MCM-48 was prepared using 20 wt.% [Emim][Gly].

### 2.3. Characterization

Thermogravimetric analysis (TGA) of all of the samples was performed using a TGA-50 instrument manufactured by Shimadzu (Tokyo, Japan). During all of the experiments, a nitrogen flow of 50 mL per minute was maintained, while the temperature increase was set at 10 °C per minute up to a maximum of 800 °C. An estimated 10–12 mg of substance was utilized for each sample. An X-ray diffractometer (Rigaku Ultima IV, Tokyo, Japan) equipped with a Cu source with a wavelength of 1.54056 Å allowed for the detection of the crystal structure of the pristine MCM-48 support and AAILs@MCM-48 composites. The analysis was conducted over 2θ values ranging from 0.5 to 10° at a scanning rate of 1.2°/min. The N_2_ adsorption–desorption isotherms of MCM-48 and the composites were obtained using the Micromeritics ASAP (Norcross, GA, USA) instrument at 77 K (liquid N_2_). The textural properties such as specific Brunauer–Emmett–Teller (BET) and Langmuir surface area, and pore volume for each sample were computed from the corresponding N_2_ adsorption–desorption data.

### 2.4. Adsorption Isotherms

N_2_ and CO_2_ isotherms were measured utilizing a high-precision intelligent gravimetric analyzer (IGA, HidenIsochema Ltd., Warrington, UK). Utilizing the electro-balance principle, the IGA is a completely automated computer-controlled microbalance capable of measuring weights with 1 μg accuracy. A stainless-steel container containing a known-weight sample is suspended from a gold chain in one arm, while a reference weight is affixed to the other arm. The sample chamber is equipped with a pressure transducer (Druck PDCR4010, Leicester, UK, ±0.008 bar) and a thermocouple (±0.05 K) for temperature measurement. The current investigation involved the measurement of CO_2_ adsorption uptake at 303, 313, and 323 K, while the N_2_ adsorption uptake was assessed at 313 K. The isotherm measurements were performed for the pressure range from 0.1 to 10 bar. Each isotherm included a quantity of material ranging from 50 to 70 mg. The sample chamber was heated to a temperature of 453 K using a water–glycol bath and evacuated to a pressure of 10 mbar using a vacuum system (Pfeiffer, Toronto, ON, Canada) until the sample weight remained constant for 1 h. This confirmed that the solvent, moisture, and pollutants were eliminated. Following the outgassing process, the sample was brought to the desired experimental temperature by altering the temperature of the water bath. Sufficient time was allowed for the sample to reach a stable temperature. Pressure levels were preset from 0.1 to 10 bar in the IGASwin v.1.03.84 program (HidenIsochema Ltd., Warrington, UK) and isotherm measurements began when the sample was ready. A mass flow controller (MFC) controlled CO_2_ or N_2_ injection into the chamber to maintain pressure. The IGASwin program provided real-time monitoring of the mass, temperature, and pressure. After at least two hours, which allowed the pressure to reach equilibrium, the MFC introduced more CO_2_ or N_2_ at the subsequent pressure level. This was carried out for each predetermined pressure at a specified temperature. Real-time adsorption data were adjusted for buoyancy after the experiment.

## 3. Results and Discussion

### 3.1. Characterizations of the AAIL-Impregnated Sorbents

The samples were heated up to 1073 K with an N_2_ flow rate of 50 mL·min^−1^ to evaluate the thermal stability of pure [Emim][Gly], [Emim][Ala], MCM-48, and all AAILs@MCM-48 compounds. The resultant thermograms are displayed in Figure 1. The thermogram of pristine MCM-48 indicates that there is a small weight loss of 1% below 373 K, as shown in the derivative TGA (DrTGA) profile (Figure 1), and only 2% additional loss over the temperature range of 1073 K. The fact that it remains intact at temperatures up to 1073 K shows that pure MCM-48 is thermally stable. This agrees with previous studies as well, such as the work reported by Schumacher et al., who found that MCM-48 maintains its structural integrity up to 1123 K [27]. The thermograms of pristine AAILs [Emim][Gly] and [Emim][Ala] showed a small weight loss of 1 to 3% below 373 K, which can be ascribed to the moisture content, and were stable up to 473 K. The AAILs showed a dramatic decline in weight above 473 K, suggesting a quick decomposition; based on the DrTGA profile, we can estimate that the onset decomposition temperatures (T_onset_) of [Emim][Gly] and [Emim][Ala] are around 488 and 498 K, respectively. When heated to 1073 K, both AAILs evaporated. Any residual solvent (methanol), physically adsorbed moisture, or other contaminants might explain why all of the composites showed a weight loss at temperatures below 373 K. As seen in the DrTGA profiles (Figure 1b,d), the composites began to lose weight at a significant rate at the T_onset_ of the pristine AAILs for temperatures between 473 and 673 K, as anticipated. Beyond 673 K, there was very little weight loss until 1073 K. The resulting weight reduction for the composites is likely due to the AAILs. Since pristine MCM-48 showed very little weight loss up to 1073 K, any weight loss is likely attributable to the impregnated AAILs. It can therefore be concluded that the composites are thermally stable and that the thermograms show that the AAILs have been successfully impregnated.

To elucidate the impact of incorporated AAILs on the MCM-48 support structure, both pristine MCM-48 and AAILs@MCM-48 composites were analyzed using the XRD instrument. The analysis was performed within an angular range from 0.5° to 10°, with a scanning rate of 1.2° per minute. The resulting diffractogram is displayed in Figure 2. The unaltered MCM-48 exhibited prominent characteristic peaks at 2θ = 2.61° and a weak reflection peak at 4.5°, which aligns well with the findings in the existing literature [8,28]. After the addition of [Emim][Gly] and [Emim][Ala], the composites exhibited a consistent peak at around 2θ = 2.61° for all of the various loadings. However, a subsequent decline in the peak intensity of the primary characteristic peak was detected with a higher AAIL loading. Additionally, the peak at higher indices also vanished. It was also observed that there was a slight shift in peaks upon impregnation of [Emim][Ala] in MCM-48. Kim et al. also reported a similar decrease in intensity and a minor shift in the characteristic peak for aminopropyl-attached MCM-48 [3] which was attributed to pore filling by aminopropyl groups and interaction with the support. Similar observations were made for PEI impregnated in MCM-41 [29]. The XRD patterns of the composites demonstrate that the MCM-48 solid support structure remains unchanged during the impregnation procedure.

To unfold the textural properties of the composites upon encapsulation of the AAILs into the pore of the MCM support, N_2_ adsorption–desorption isotherms of MCM-48 and AAILs@MCM-48 composites were obtained at a liquid nitrogen temperature of 77 K. The results are displayed in Figure 3. From the corresponding N_2_ isotherm data, textural properties such as specific Brunauer–Emmett–Teller (BET) and Langmuir surface area, and pore volume for each sample were calculated and are reported in Table 2. In addition, the pore size distribution for pristine MCM-48 and composites was calculated using the Barett–Joyner–Halenda (BJH) model and is presented in Figure 4. The unmodified MCM-48 support displayed typical type IV reversible isotherm characteristics of mesoporous material, with sharp steps for the pressure range from P/P_0_ = 0.2 to 0.3 associated with the capillary condensation and without noticeable hysteresis. A similar isotherm for MCM-48 is also reported in previous literature reports [3,27,30]. The composites displayed significantly reduced N_2_ adsorption compared to the original MCM-48. The isotherms also indicated the presence of hysteresis between P/P_0_ values of 0.5 and 0.9, which may be ascribed to the filling of the pores and the blockage of the pore network by the encapsulated AAILs. According to Table 2, increasing the loading of AAILs leads to a further drop in both surface area and pore volume. For example, the BET surface area of composites 40-[Emim][Gly]@MCM-48 and 40-[Emim][Ala]@MCM-48 decreased to 50 and 29 m^2^ g^−1^, respectively, and the pore volume decreased to 0.07 and 0.04 cm^3^ g^−1^. This suggests that when a large amount of AAILs is loaded, the pores of the MCM-48 support are essentially occupied by the enclosed AAILs, which is also evident from the pore size distribution.

### 3.2. CO_2_ Adsorption Isotherms

CO_2_ absorption capacities were measured at 303, 313, and 323 K and pressures ranging from 0.1 to 10 bar for pristine MCM-48, [Emim][Gly] @MCM-48, and [Em-im][Ala]@MCM-48 composites. Figure 5b,d, f illustrate the outcomes for the low-pressure range (0.1 to 1.0 bar), whereas Figure 5a,c,e display the findings for the complete range of pressures (0.1 to 10 bar). CO_2_ uptake of pristine MCM-48 was 0.07 and 0.14 mmol g^−1^ for 0.1 and 0.2 bar at 303 K, respectively. CO_2_ uptake increased linearly with the increase in pressure, indicating that the process followed physisorption. The [Emim][Gly]-incorporated composite [Emim][Gly]@MCM-48 exhibited enhanced CO_2_ uptake compared to pristine MCM-48 at a pressure below 2 bar. As an example, the CO_2_ adsorption capacity for the 20%-[Emim][Gly]@MCM-48 sample was 0.19 mmol g^−1^ at 0.2 bar and 303 K. CO_2_ uptake increased further with the increase in the loading at lower pressure. For 40 wt.% loading of [Emim][Gly], the composite exhibited CO_2_ uptakes of 0.74 and 0.82 mmol g^−1^ for 0.1 and 0.2 bar at 303 K, respectively, values that are quite high and are 10- and 5-fold greater than pristine MCM-48 under the same conditions.

A similar surge in CO_2_ uptake was observed for [Emim][Ala] composites, where CO_2_ uptake increased with the increase in [Emim][Ala] loading (Figure 6). The 40 wt.% loaded [Emim][Ala]@MCM-48 exhibited the highest CO_2_ adsorption rate of 0.65 and 0.74 mmol g^−1^ at 0.1 and 0.2 bar at 303 K, respectively, which is a 9- and 5-fold increase relative to pristine MCM-48. It is important to highlight that under identical temperatures and pressures, the CO_2_ adsorption capacity of the [Emim][Ala]@MCM-48 composites was marginally lower than that of the [Emim][Gly]@MCM-48 composites with the same AAIL loading.

The substantial increase in the CO_2_ sorption capacity of the AAILs@MCM-48 sorbent following AAIL impregnation under post-combustion conditions (P_CO2_ ≈ 0.15 bar) can be attributed to CO_2_’s strong affinity for the amino group attached to the anion of the [Emim][Gly] and [Emim][Ala] that were introduced into the pore. According to published research [17,31,32], the amino group of AAILs reacts with CO_2_ via a mechanism analogous to that of aqueous amine solution. According to Wang et al. [24], the reduced size of the cation and anion of [Emim][Gly] enables them to approach an amino group and undergo a reaction resulting in the formation of carbamate with a stoichiometry of 1:2 (Figure 1). Hence, it can be hypothesized that the amino groups present in composites of [Emim][Gly]@MCM-48 and [Emim][Ala]@MCM-48 engage in comparable interactions, culminating in the formation of carbamate via a reaction with CO_2_. This process enhances the CO_2_ uptake capacity of [Emim][Ala]@MCM-48 relative to its pristine counterpart below a pressure of 1 bar.

Hence, although surface area and pore volume diminished significantly upon encapsulation of [Emim][Gly] and [Emim][Ala], they provide the chemically active sites to attract CO_2_. As a result, at lower pressure, chemisorption acts as a dominating factor. However, as the pressure increases, there are fewer chemical active sites available as they are already occupied, so the available surface area and pore volume play a dominant role at a pressure of 2 bar, and the higher the loading of AAILs, the lower the available active surface. As the pressure rises to moderate and high levels, the adsorption capacity of the sorbent is determined not only by the active chemical adsorption sites inside the sorbent but also by the physical adsorption sites that are present [33]. Hence, as observed from Figure 5 and Figure 6, at a pressure above 2 bar, CO_2_ uptake was lower for all AAIls@MCM-48 composites compared to pristine MCM-48 at the three temperatures studied. CO_2_ uptake decreased across the board for all composites when the temperature ramped up to 323 K from 303 K at the same pressure.

Table 3 presents the CO_2_ uptake by various amines and ionic liquid-functionalized silica-based solid sorbents such as MCM-48, MCM-41, and SBA reported in the literature along with our present study. Functionalized silica performed better than pristine silica. Kim et al. [3] functionalized MCM-48 with four different monomeric and polymeric hindered and unhindered amines such as aminopropyl (APS), polyethyleneimine (PEI), pyrrolidinepropyl (PyrPs), and polymerized aminopropyl (p-APS). It was revealed that among all four composites, APS-MCM-48 demonstrated the highest CO_2_ adsorption capacity, which was 0.8 mmol g^−1^, and p-APS-MCM-48 performed the worst, with 0.1 mmol g^−1^, under the same experimental condition of 298 K and 1 atm. They attributed the differences to the concentration of surface amino groups, the strength of CO_2_–amine interactions, and the accessibility of amino groups. PEI-MCM-41 demonstrated a notably higher CO_2_ uptake of 1.09 mmol g^−1^ at 101 kPa and 298 K compared to APS-MCM-48. Additionally, the CO_2_ uptake of [P666][2-Op]-MCM-41 was slightly higher at 1.17 mmol g^−1^ under similar experimental conditions as PEI-MCM-41 (101 kPa/298 K). The present study demonstrates notable improvements in CO_2_ capture efficiency compared to previous research efforts at much lower pressures. For instance, the composite material 40-[Emim][Gly]-MCM-48 exhibited a CO_2_ uptake of 0.82 mmol g^−1^ at 20 kPa and 303 K; moreover, at a higher pressure of 100 kPa and the same temperature, the CO_2_ uptake for 40-[Emim][Gly]-MCM-48 increased to 1.15 mmol g^−1^, demonstrating better performance compared to other materials tested under similar conditions. These findings highlight the effectiveness of the present study’s composite materials in enhancing CO_2_ capture capabilities, offering promising prospects for advancing carbon capture technology.
nanomaterials-14-00514-t003_Table 3Table 3CO_2_ uptake by various amine/IL-modified silica-based composites reported in the literature.CompositesCO_2_ Uptake(mmol g^−1^)Experimental ConditionsRef.APS-MCM-480.81.0 atm/298 K[3]PEI-MCM-480.41.0 atm/298 K[3]PyrPS-MCM-480.31.0 atm/298 K[3]p-APS-MCM-480.11.0 atm/298 K[3]35PEHA-15DEA-MCM-480.511.0 atm/298 K[34]50PEHA-MCM-480.261.0 atm/298 K[34]PEI-MCM-411.09101 kPa/298 K[35][P666][2-Op]-MCM-411.17101 kPa/298 K[36][P4444][imidazole]-MCM-410.91101 kPa/293 K[37]EDA-PVC-SBA-150.550 kPa/298 K[38]40 wt.%-[Emim][Gly]-MCM-480.8220 kPa/303 KPresent Study40 wt.%-[Emim][Ala]-MCM-480.7420 kPa/303 KPresent Study40 wt.%-[Emim][Gly]-MCM-481.15100 kPa/303 KPresent Study40 wt.%-[Emim][Ala]-MCM-481.1100 kPa/303 KPresent Study


### 3.3. Selectivity for CO_2_/N_2_

To be an effective solid sorbent in the post-combustion capture process, it is crucial to have excellent selectivity of CO_2_ over other gases, particularly N_2_. Therefore, to determine the selectivity of CO_2_/N_2_, we conducted measurements of N_2_ adsorption isotherms at 40 °C. The pressure range for each isotherm spanned from 0.1 to 10 bar. The optimal selectivity may be determined by many approaches, one of which involves computing the selectivity based on the isotherms of individual components. This technique relies on the adsorption of the components at identical pressure, as shown in Equation (1) [39].
(1)$$S=\frac{q_{CO2}}{q_{N2}}$$
where S denotes the selectivity, and q_CO2_ and q_N2_ symbolize the adsorption of CO_2_ and N_2_ in mole, respectively. The computed ideal CO_2_/N_2_ selectivity from the isotherms of CO_2_ and N_2_ at 313 K of [Emim][Gly]@MCM-48 and [Emim][Ala]@MCM-48 composites are displayed in Figure 7. It was found that pristine MCM-48 exhibited almost constant CO_2_/N_2_ selectivity of about 2 for the entire pressure range. Meanwhile, [Emim] [Gly]- and [Emim][Ala]-encapsulated MCM-48 composites displayed higher CO_2_/N_2_ selectivity at lower pressures below 2 bar and their selectivity increased with the increase in loading. Out of all of the composites containing [Emim][Gly]@MCM-48, the 40 wt.%-[Emim] [Gly]@MCM-48 composite exhibited the best selectivity of 11 and 8 at pressures of 0.1 and 0.2 bar, respectively. However, the selectivity steadily decreased as the pressure increased. Likewise, 40 wt.%-[Emim][Ala]@MCM-48 exhibited the highest selectivity among all [Emim][Ala]@MCM-48 composites, reaching 17 and 11 at 0.1 and 0.2 bar, respectively.

The significant surge in CO_2_/N_2_ selectivity for the composites compared to pristine MCM-48 can be attributed to the presence of encapsulated amino acid-based liquid in the pore of MCM-48. As stated earlier, the loading of AAILs resulted in a significant increase in the amount of CO_2_ that was taken in. This may be due to the active chemical sorption sites for carbon dioxide provided by the amino acids. It is hypothesized that these sites generate an N-C bond that is similar to the interaction that occurs between CO_2_ and alkanolamine. Even though the occupied ionic liquid resulted in a decrease in the surface area and pore volume, the amount of CO_2_ captured was dramatically enhanced. On the other hand, owing to the physical nature of adsorption, which is reliant on the available surface area, N_2_ does not have any affinity for the amino group and was therefore not adsorbed. The outcome is a higher CO_2_/N_2_ selectivity because at low pressure, the absorption of carbon dioxide is more prevalent than the uptake of nitrogen. However, as the pressure is increased, the adsorption capacity is also governed by the physical adsorption sites present. This is in addition to the active chemical adsorption sites that are present in the sorbent. As a consequence, the selectivity of the composites for CO_2_/N_2_ diminishes as the pressure increases, and it reaches a level that is lower than that of pristine MCM-48 when the pressure is more than 2 bar.

A closer look at both AAIL composites reveals that the 40 wt.%-[Emim][Ala]@MCM-48 exhibited higher CO_2_/N_2_ selectivity than the 40 wt.%-[Emim][Gly]@MCM-48, although the CO_2_ uptake of 40 wt.%-[Emim][Gly]@MCM-48 was higher than that of 40 wt.%-[Emim][Ala]@MCM-48. This can be attributed to the lower N_2_ uptake by 40 wt.%-[Emim][Ala]@MCM-48, with a lower surface area and pore volume available, compared to 40 wt.%-[Emim][Gly]@MCM-48, as explained earlier. Hence, despite the lower CO_2_ uptake by 40 wt.%-[Emim][Ala]@MCM-48, CO_2_/N_2_ selectivity was higher.

### 3.4. Equilibrium Isotherm Modeling

Equilibrium isotherm modeling is crucial for accurately representing the experimental data in the design of adsorption and desorption processes. The composite in this work exhibits both robust and weak binding sites as a result of the inclusion of encapsulated AAILs inside the pore of MCM-48. Therefore, after considering many models, the Dual-Site Langmuir model (DSL) [14,40] was determined to be appropriate. This model integrates Langmuir adsorption at two different sites, and the overall adsorption is the sum of the adsorption at each site, as shown in Equation (2) [41]:(2)$$N_{e}=\frac{N_{A}b_{A}P}{1+b_{A}P}+\frac{N_{B}b_{B}P}{1+b_{B}P}$$
where N_e_ represents CO_2_ uptake (mmol·g^−1^), *P* represents the pressure (bar), and *N_A_*, *N_B_*, *b_A_*, and *b_B_* represent the DSL parameters. The regressed parameters for the composites [Emim][Gly]@MCM-48 and [Emim][Ala]@MCM-48 are, respectively, presented in Table 4 and Table 5. Figure 8 and Figure 9 depict the fitting contours of the DSL model. The model provided an excellent fit to the experimental data, as indicated by the proximity of the R^2^ values to unity. Consequently, the enthalpy of adsorption will be calculated using the model data in the following section.

### 3.5. Isosteric Heat of Adsorption (Q_st_)

The isosteric heat of adsorption (adsorption enthalpy, *Q_st_*) is a crucial metric in the adsorption-based CO_2_ capture process. It discloses the attraction and intensity of the interaction between the gas molecules being absorbed and the host. Therefore, it signifies the magnitude of the energy needed for the adsorption–desorption process. The adsorption enthalpy (*Q_st_*) was calculated based on the CO_2_ isotherms obtained at 303, 313, and 323 K. The DSL model was employed to initially fit the isotherms, as mentioned in the previous section. Following this, the Clausius–Clapeyron equation, represented by Equation (3), was used [41].
(3)ln⁡PN=−QstR1T+C
where *P* stands for pressure (bar), *N* is CO_2_ uptake, *R* is the universal gas constant, and *T* is temperature (K). According to the equation, the graphs of (ln *P*) vs. 1/*T* were plotted for a constant CO_2_ uptake. The slope of the plots represents the *Q_st_* corresponding to the CO_2_ uptake. The calculated *Q_st_* for pristine MCM-48 and the AAILs@ MCM-48 composites are illustrated in Figure 10. The *Q_st_* values for pristine MCM-48 were approximately 20 kJ mol^−1^, which also confirms the physisorption nature of CO_2_ adsorption, whereas a sharp increase in *Q_st_* was observed for [Emim][Gly] and [Emim][Ala] incorporated into the MCM-48 composites. An upward trend in the isosteric heat of adsorption was noted as the concentration of AAILs on both sorbents increased incrementally. The maximum values of *Q_st_* were −71 and −77 kJ mol^−1^, respectively, for 40 wt.%-[Emim][Gly]@MCM-48 at 0.7 mmol g^−1^ CO_2_ uptake and 0.6 mmol g^−1^ CO_2_ uptake, respectively. The *Q_st_* values reached their maximum at low pressures ranging from 0.1 to 0.2 bar, at the initial stage of CO_2_ adsorption. The observed rise in *Q_st_* can be ascribed to a substantial increase in CO_2_ adsorption within the low pressure range, which leads to a greater release of heat throughout the adsorption process. As previously described, the amino group of the anion of the TSIL is capable of forming an N–C bond and, as a result, liberating a greater quantity of energy during CO_2_ adsorption [41]. With additional CO_2_ adsorbed, the value of *Q_st_* decreases significantly after the peak, reducing the number of amine sites with the highest affinity that are available for CO_2_ molecules to occupy. Comparable trends in *Q_st_* variations were detected upon incorporating [Bmim][Ac] and [Emim][Ac] into ZIF-8, consistent with findings reported in a previous investigation conducted by our research group [40].

## 4. Conclusions

The objective of this work was to enhance the capacity for capturing CO_2_ and the selectivity of CO_2_ over N_2_ by encapsulating amino acid-based ionic liquids (AAILs) [Emim][Gly] and [Emim][Ala] into ordered mesoporous silica MCM-48, resulting in the formation of AAILs@MCM-48 composites. The thermogravimetric and XRD characterizations of the composites indicate that the thermal and structural integrity of the original MCM-48 support remained unchanged after the AAILs were encapsulated. An N_2_ adsorption–desorption analysis conducted at 77 K revealed a substantial decrease in surface area and pore volume as the support’s pores were almost filled with increasing AAIL loading. This finding further verifies the effective encapsulation of AAILs inside the porous support. Both [Emim][Gly]@MCM-48 and [Emim][Ala]@MCM-48 showed enhanced CO_2_ adsorption compared to the unmodified MCM-48 under post-combustion flue gas conditions, with a CO_2_ partial pressure of around 0.15 bar. Among the composites, the 40 wt.%-[Emim][Gly] composite showed the maximum CO_2_ uptake of 0.74 and 0.82 mmol g^−1^ at 0.1 and 0.2 bar, respectively, at 303 K. These values represent an increase of nearly 10- and 5-fold compared to pristine MCM-48 under the same conditions. The composites demonstrated both enhanced CO_2_ absorption and increased CO_2_/N_2_ selectivity. The selectivity of the 40 wt.% [Emim][Ala]@MCM-48 composites significantly increased to 17 at 0.1 bar, while the selectivity of pristine MCM-48 was only 2. The presented new composites outperform other silica-based composites reported in the literature, even those using amines as solvents. It can therefore be inferred that AAILs@MCM-48 composites possess significant potential to be considered candidates for post-combustion CO_2_ capture.

## Data Availability

The data presented in this study are available on request from the corresponding author.

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
