# Peer review of "Functionalization of Ordered Mesoporous Silica (MCM-48) with Task-Specific Ionic Liquid for Enhanced Carbon Capture"

_nanomaterials, 2024, doi:10.3390/nano14060514_

Round 1

Reviewer 1 Report

Comments and Suggestions for Authors

This paper presents noble composites of AAILs@MCM-48 by functionalizing ordered mesoporous silicon MCM-48 with two amino acid-based ionic liquids (AAILs) ([Emim][Gly] and 10 [Emim][Ala]) to improve carbon capture and selectivity of CO2 over nitrogen. The topic of the paper is adequate for the journal. Also, this paper is well structured, and important previous studies are well summarized. The result is easy to follow even though more detailed explanations and justifications are necessary. I recommend publishing this paper when the following issues are addressed

1. In lines 179 and 182, “Error! Reference source not found” can be found in the manuscript.

2. In line 201, it should be 77K instead of 77.

3. In the Introduction, please provide a discussion on the use of ionic liquids in the field of CO2 capture, with a focus on highlighting the significant role of ionic liquids in carbon capture. Please refer to the literature (Energy 113(2016) 1-8; Energy 87(2015) 165-172; Energy 161 (2018) 1122-1132).

4. Please provide more intuitive and specific information on AAIL@MCM-48 preparation conditions, such as temperature, speed, time, etc.

5. Please compare the adsorption performance with similar literature to better illustrate the advantages of the adsorbent studied in this article.

6. The CO2 adsorption efficiency of [Emim] [Gly] @ MCM-48 is higher than that of [Emim] [Ala] @ MCM-48. But the latter has better selectivity. Although the author has explained, can it provide strong literature support?

7. The author chose the concentration range of AAIL as 20,30,40. From the results in the article, it can be seen that at lower partial pressures, the higher the concentration, the better the adsorption effect. Does the author consider the adsorption effect of the adsorbent when the concentration of AAILs is above 40%, such as 60%?

Author Response

Please check it in attachment.

Reviewer 2 Report

Comments and Suggestions for Authors

This article entitled “Functionalization of Ordered Mesoporous Silica (MCM-48) with Task Specific Ionic Liquids to Enhance Post-combustion CO2 Capture Capacity” describe an efficient procedure to encapsulate two amino acid-based ionic liquids (AAILs), ([Emim][Gly] and [Em-im][Ala]) on suport MCM-48 in order  to improve carbon capture and selectivity of CO2 over nitrogen.

 The resulting materials  [Emim][Gly]@MCM-48 and [Emim][Ala]@MCM-48 demonstrated improved CO2 adsorption in comparison to unmodified MCM-48, with a CO2 partial pressure of around 0.15 bar.

This manuscript is well organised and supported. 

Minor revisions:

1. The stability of the composites in the CO2 capture operation was investigated by performing multiple cycles of adsorption at 313 K and desorption at 373 K at the atmospheric pressure in the presence of N2 for  [Emim][Gly]@MCM-48 and [Emim][Ala]@MCM-48?
If yes, the results showed that the composite sorbent could almost maintain its original adsorption capacity during multicycle operations? Please attach the corresponding graph.
Could the composites be readily regenerated?

2.  I request the incorporation of scanning electron microscopy (SEM) images into the manuscript to analyse the morphological changes that occurred in the material when the AAILs were embedded within the MCM-48 support.

The analyzes performed, namely the thermogravimetric and XRD characterization of the composites indicate that the thermal and structural integrity of the original MCM-48 support remained unchanged after the encapsulation of AAILs.
Therefore, it can be inferred that AAILs@MCM-48 composites possess significant potential to be considered as good candidates for post-combustion CO2 capture.

Author Response

Please check it in attachment.

Reviewer 3 Report

Comments and Suggestions for Authors

This paper presented AAILs@MCM-48 composites which were used to improve carbon capture and selectivity of CO2. Overall, this is an interesting work. However, some issues should be further considered before acceptance.

1.      In Figure 1a, there are two TG curves for the same sample (20-[Emim][Gly]MCM-48). Please double check it.

2.      The characteristically peaks and corresponding functional groups should be marked in the FTIR patterns.

3.      In Figure 2, there is an obvious peak shift for Ala-modified sample after compositing, why?

4.      Will amino functional groups (-NH2) be oxidized during the compositing process? XPS should be performed to confirm this.

5.      What is the pore size distributions for the samples before and after composition?

6.      Some related works should be cited to improve the paper: e.g., Adv. Powder Mater. 2022, 1, 100018; Chemical Engineering Journal, 2022, 449, 137561; 

Comments on the Quality of English Language

Minor editing of English language required.

Author Response

Please check it in attachment.

Reviewer 4 Report

Comments and Suggestions for Authors

The current article is devoted to the formation of composites made of ionic liquids and MCM-48 and their application for CO2 capture. The current version of the article looks unread. The authors should pay more attention on the content and English. For example:

-Please check chemical formulas, use subscript for indexes. Line 16: "with a CO2 partial pressure of around 0.15 bar", line 250: "attract CO2"

-Line 38 "is known to have has". "Thermogravimetric analysis (TGA) of all the samples was performed using a TGA-50 instrument manufactured by Shimadzu was utilized"

-Lines 178: "which are displayed in Error! Reference source not found.."

Moreover, the introduction is badly written, it does not provide clear introduction into the topic. Please add more details.

-Why were these 2 ion liquids used? Which are alternatives?

Comments on the Quality of English Language

The article must be clearly checked before publication as it contains numerous mistakes.

Author Response

Please check it in attachment.

Round 2

Reviewer 3 Report

Comments and Suggestions for Authors

I think the revised paper could be published in current form. 

Reviewer 4 Report

Comments and Suggestions for Authors

The article can be published in the current version.

Comments on the Quality of English Language

The article can be published in the current version.